# Discontinuous rate-stiffening in a granular composite modeled after cornstarch and water

David Z. Chen[1], Hu Zheng[1,2,3], Dong Wang[1] & Robert P. Behringer[1]

Cornstarch in water exhibits impact-activated solidification (IAS) and strong discontinuous shear thickening, with "shear jamming". However, these phenomena are absent in cornstarch in ethanol. Here we show that cornstarch granules swell under ambient conditions. We postulate that this granule swelling is linked to an interparticle force scale that introduces a discontinuous rate-dependence to the generation of stable contacts between granules. We studied this force scale by coating sand with ~2 μm-thick polydimethysiloxane, creating a material that exhibits a similar IAS and discontinuous deformation rate-stiffening despite being a granular composite, not a suspension. This result suggests rate-dependence can be tuned by coating granular materials, introducing an interparticle force scale from rate-dependent properties present in the coating material. Our work provides insights into the unique behavior of cornstarch in water, bridges our understanding of suspensions and dry granular materials, and introduces a method to make discontinuous rate-dependent materials without suspending particles.

[1] Department of Physics and Center for Nonlinear and Complex Systems, Duke University, Durham, NC 27708, USA. [2] Department of Geotechnical Engineering, College of Civil Engineering, Tongji University, Shanghai 200092, China. [3] School of Earth Science and Engineering, Hohai University, Nanjing Jiangsu 211100, China. Correspondence and requests for materials should be addressed to H.Z. (email: tjzhenghu@gmail.com)

Cornstarch in water shows non-Newtonian behavior: impact-activated solidification (IAS), where a high-speed intruder is stopped by the rapid solidification of the underlying suspension[1], and discontinuous shear thickening (DST), where increasing shear rates leads to a sudden increase in viscosity that spans several orders of magnitude[2]. This type of non-Newtonian response can be highly desirable, in the case of flexible body armor[3,4], or highly undesirable, such as in industrial processes with confining flow, where increasing flow rates constricts the flow[5]. Despite a number of studies on the properties of cornstarch in water, the underlying mechanism for its unusually strong response under shear compared to typical shear thickening suspensions[6,7] and "shear jamming" response[8,9] has been elusive: hydrodynamic forces[10,11], dilatancy[12,13], and shear jamming[14,15] have all been suggested to play a role in DST. This lack of understanding of the underlying mechanisms involved makes it prohibitive to explore the material design space in order to make new materials with strongly discontinuous rate-dependent properties. In addition, it is difficult to take effective measures to prevent DST in suspensions when it is not desired.

Here we show that IAS and DST in aqueous cornstarch suspensions are strongly influenced by a previously unexplored phenomenon, namely that cornstarch swells appreciably in water under ambient conditions, which alters its mechanical properties. We postulate that this swelling introduces a granule-level force scale that (1) allows for strong frictional contacts to form, and (2) leads to a discontinuous deformation rate-dependence on stiffness/viscosity via a granule-level force scale. We modeled this granule-level force scale using fine, ~ 60–250 μm-diameter, sand coated with a ~ 2 μm-thick layer of polydimethylsiloxane (PDMS), which gives rise to a rate-dependent force scale modulated by the viscoelastic properties of PDMS. This model PDMS-coated sand exhibits qualitatively similar behavior to cornstarch and water, namely impact-activated solidification/ jamming and discontinuous-like deformation rate-stiffening. However, it remains distinctly a dry granular material rather than a suspension. Our finding gives insight into IAS and DST in suspensions, suggests similarities between suspensions and dry granular materials, and provides a blueprint for creating strongly deformation rate-dependent materials with no suspending fluid.

## Results

**Swelling of cornstarch in water**. Cornstarch in water is a classic discontinuously shear thickening suspension. However, despite its popularity and ubiquity, we find that it has unique properties. For example, cornstarch swells in water under ambient conditions (see Supplementary Information for video). Under optical microscopy, we observed a ~ 12% swelling in diameter or ~ 40% by volume (Fig. 1). This swelling is unobserved for cornstarch in ethanol, a slightly weaker polar protic solvent compared to water (dipole moment: 1.69 Debye versus 1.85 Debye). We observed that adding cornstarch to water produces an exothermic reaction, releasing ~ 45.3 J/g of cornstarch in 100 ml of water. In contrast, adding cornstarch to the same volume of ethanol is endothermic, consuming ~ 24.2 J/g of cornstarch (see Supplementary Information for video). This suggests that swelling of cornstarch is energetically favorable in water and unfavorable in ethanol, which explains the absence of swelling in cornstarch–ethanol at room temperatures. To quantify this swelling in terms of a change in overall volume fraction of cornstarch, we repeatedly mixed cornstarch with water inside of a graduated cylinder and observed a total volume increase of ~11 ± 2% (of cornstarch volume before swelling), instead of ~ 40% (see Methods for details). This means that the cornstarch granules absorb some of the surrounding water during swelling, ultimately resulting in a volume fraction

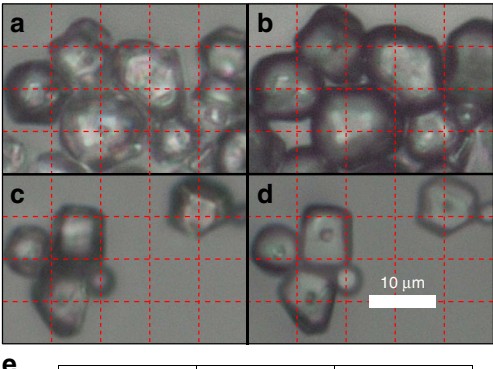

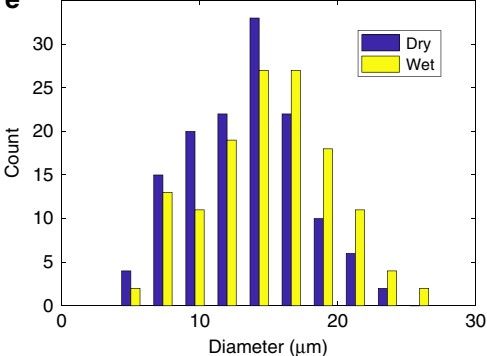

**Fig. 1** Optical microscopy and particle size measurements. **a** Image of native cornstarch granules before adding water. **b** Image of cornstarch granules after adding water. Particles swell by ~12% in diameter, or ~ 40% by volume on average. **c** Image of cornstarch before adding ethanol. **d** Image of cornstarch after adding ethanol. No observable swelling occurs in ethanol. Cornstarch granules are polydisperse, with average diameters in the range ~10–40 μm. **e** Histograms of particle sizes measured with water (wet) and without (dry)

increase of ~11%. One previous study has found that cornstarch in water is porous, which is in agreement with our observation of cornstarch swelling[16]. These findings suggest two unique properties of cornstarch–water suspensions: (1) granule swelling increases the actual packing fraction of the cornstarch–water suspension by ~11%, and (2) this swelling may alter the mechanical properties of cornstarch granules. Due to this swelling effect of cornstarch in water, henceforth we denote the volume fraction before swelling as $\varphi_{nom}$, and denote the packing fraction calculated accounting for swelling as $\varphi_{act}$.

**Impact experiments on cornstarch suspensions**. To test the effect of cornstarch swelling on mechanical properties, we performed impact experiments on cornstarch and water mixture, and cornstarch and ethanol mixture, which showed that cornstarch and ethanol mixture loses its shear thickening behavior (Fig. 2). We dropped a 6.35-cm-diameter disk intruder (thickness: 1.1 cm and mass: 291 g) into cornstarch–water and cornstarch–ethanol mixtures ($\varphi_{nom}$ ~ 45% for both), and tracked its depth and velocity as a function of time (see Supplementary Figures 1 and 2 and Supplementary Note 1 for setup). For the cornstarch aqueous suspension, the intruder impacted the surface at ~ 3 m/s and rebounded, indicating that the suspension became solid-like and reacted elastically to the impact. In contrast, the intruder impacting into the ethanol solution encountered minimal resistance, quickly sinking to the bottom of the container. Increasing cornstarch–ethanol packing fraction to $\varphi_{act}$ ~ 54% does not resolve this qualitative difference in response to impacts, nor does accounting for granule swelling (Fig. 2b). In every case that we have accounted for, i.e., $\varphi_{nom}$ from 43 to 47% for

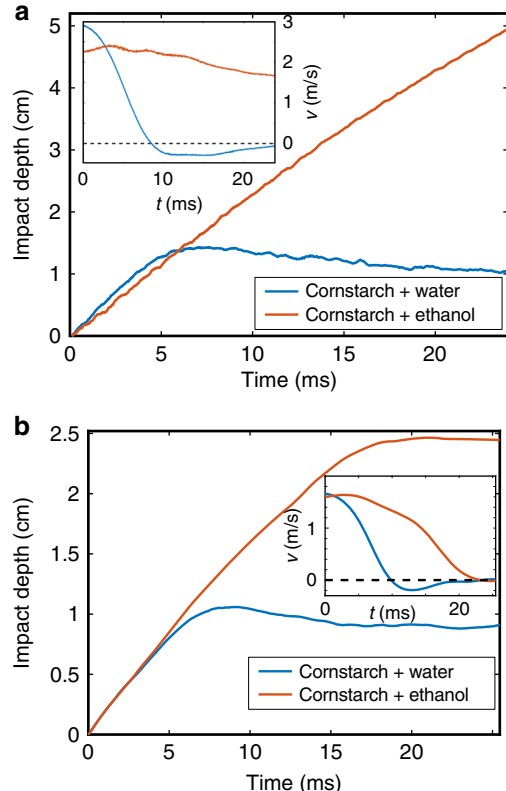

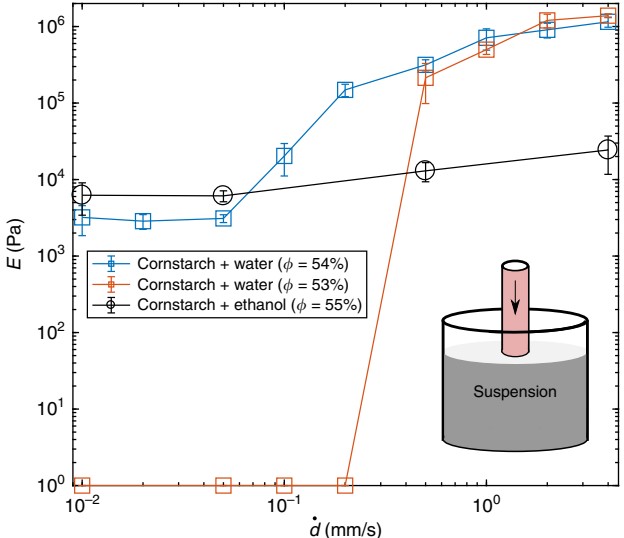

**Fig. 3** Mechanical indentation experiments for dense cornstarch packings in water and in ethanol. The elastic modulus versus displacement rate shows discontinuous stiffening for cornstarch in water, which transitions from fluid-like behavior (zero stiffness) below 0.5 mm/s to solid-like behavior at $\varphi_{act} \sim 53\%$. This is absent in cornstarch in ethanol and bare cornstarch. We calculate $E$ from contact mechanics using the indicated loads (cylindrical indenter). Error bars represent standard deviation

**Fig. 2** Impact experiments on cornstarch suspensions. **a** Impact depth vs. time for an intruder dropped into suspensions of cornstarch in water (blue) and ethanol (red), both at nominal packing fractions of $\varphi(nom) \sim 45\%$ (without accounting for swelling). **b** Impact depth vs. time for an intruder dropped into cornstarch and water mixure at $\varphi_{act} \sim 54\%$ (accounting for granule swelling), and cornstarch and ethanol mixture at $\varphi \sim 54\%$. In both cases, the cornstarch and water mixture exhibits qualitatively different behavior, rebounding the intruder on impact. The cornstarch and ethanol mixture provides minimal resistance to the intruder at $\varphi \sim 45\%$. At $\varphi \sim 54\%$, the cornstarch and ethanol mixture behaves similar to loosely packed grains surrounded by fluid, stopping the intruder after $\sim 20$ ms

cornstarch–water mixture and $\varphi_{nom} = \varphi_{act}$ from 43 to 54% for cornstarch–ethanol mixture, cornstarch in water reacts strongly to impact and is able to repel the intruder, while cornstarch in ethanol behaves qualitatively similar to a granular material in a fluid, resisting the intruder only when the packing fraction is high enough for a solid packing (i.e., greater than random loose packing (RLP), $\varphi \sim 54\%$). This highly solvent-dependent behavior suggests that cornstarch granules may have different structural properties in water compared to in ethanol, and that this structural difference is essential for the emergence of IAS and likely DST.

The swelling that we observed in cornstarch–water suspensions suggests that many previous studies on cornstarch in water likely involved higher packing fractions than the reported values, as $\varphi$ is typically estimated based on bulk densities[2,17]. This may explain why many studies have observed DST in cornstarch–water suspensions at consistently low packing fractions, below where jamming can occur. Typical cornstarch–water packing fractions, $\varphi_{nom} \sim 45$–48%[2,9], at the emergence of DST are appreciably lower than random loose packing (RLP), $\varphi \sim 55\%$[18], and random close packing (RCP), $\varphi \sim 64\%$[19]. They are also comparatively lower than the $\varphi$ of emergent DST in other suspensions, e.g., glass beads in mineral oil[2], polystyrene beads in aqueous solution[20], and polyvinyl chloride in Dinch and mineral oil[21]. Cornstarch–water

suspensions are thought to exhibit "shear jamming", where a solid-like state is reached under shear without compression[14,15]. However, granular shear jamming occurs at relatively high packing fractions, between RLP and isotropic jamming[8]. Particle packings below RLP typically do not satisfy isostaticity or rigidity percolation[18] and cannot support mechanical load through stable frictional contacts, which play a crucial role in DST, as shown in simulations[22]. Our observation of granule swelling suggests that the actual packing fraction at which DST occurs in cornstarch suspensions is likely much higher, $\varphi_{act,c} \sim 50$–53%, providing a pathway for shear jamming to occur and potentially reconciling this discrepancy.

**Rate-dependent stiffness of cornstarch–water mixture.** While the packing fraction is important in determining whether shear thickening occurs, it cannot explain the dramatic difference in behavior between cornstarch in water and cornstarch in ethanol. How does granule swelling affect the mechanical properties of cornstarch, and how does it have a dramatic effect on the rheology of cornstarch suspensions? We explored these questions by performing flat-punch indentation tests on dense suspensions/packings of cornstarch granules in water and in ethanol with packing fractions from $\varphi_{act} \sim 53$–55%, after swelling (Fig. 3, see Methods for details). With this experiment, we have precise control over the deformation rate, and we can directly measure the stiffness response of our suspensions, giving us a pathway to probe a deformation rate-response that is related to both DST and IAS.

We found that cornstarch in water exhibits a discontinuous deformation rate-dependent response, with elastic modulus increasing by several orders of magnitude over a displacement rate range of less than one order of magnitude (modulus calculated from contact mechanics, see Methods). At $\varphi_{act} \sim 53\%$ (accounting for swelling), we observed that cornstarch in water behaves like a fluid at displacement rates below $\sim 0.2$ mm/s (zero stiffness), but sharply develops a nonzero stiffness, becoming solid-like, above a displacement rate of $\sim 0.5$ mm/s. Increasing the

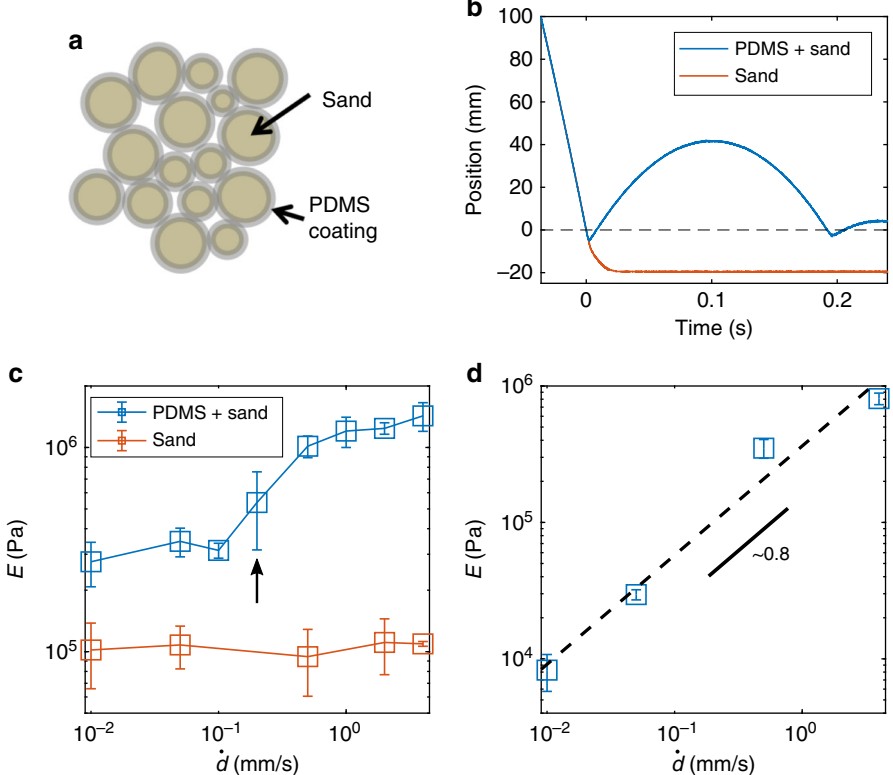

**Fig. 4** PDMS-coated sand and mechanical indentation experiments. **a** Schematic of sand particles coated with cross-linked PDMS. **b** Intruder impact position versus time into PDMS and sand and bare sand. **c** Elastic modulus versus displacement rate for coated and bare sand. The arrow indicates the critical displacement rate. **d** Elastic modulus versus displacement rate for native PDMS. PDMS-coated sand exhibits a sharp discontinuous-like stiffening response to deformation rate that is absent in bare sand. This is also absent in native PDMS, in which the stiffness increases continuously with increasing displacement rate. We attribute this substantial difference in mechanical response to the soft outer polymer layer with a stiffer granular core, which provides a force scale that is sensitive to displacement rate. Error bars in **c** and **d** represent standard deviation

packing fraction by 1% to $\varphi_{act} \sim 54\%$ (close to RLP) changes the suspension into a loose packing of cornstarch and water, where the system has a finite elastic modulus (~ 2 MPa) for displacement rates from 0.01 to 0.05 mm/s, where modulus begins to increase discontinuously with displacement rate. This strongly rate-dependent response is absent in cornstarch in ethanol, where $E$ increases by a factor of ~ 4 over more than three orders of magnitude of displacement rates. Dry cornstarch results are not shown due to limitations in force resolution, which prevented reliable results for dry cornstarch. Preliminary results indicated that dry cornstarch has no rate-dependence.

Lubrication forces present a barrier to frictional contacts in smooth and hard particles[9,21,22], but a soft (and rough) shell originating from granule swelling may provide a pathway to overcome those forces. We postulate that the mechanical changes induced by granule swelling in water contributes to its DST and IAS in suspension by introducing a force scale to the interparticle contacts: the rate at which the suspension is deformed dictates the emergence of interparticle friction. In other words, swelled cornstarch granules are repelled by lubrication at low deformation rates, but at high deformation rates, frictional contacts can form owing to granule-level changes induced by their swelling. This suggests that DST and IAS are both related to an underlying force scale, which allows for a suspension to behave fluid-like when slowly deformed and transition to solid-like at fast deformation rates. This transition to solid-like behavior may also be tied to shear jamming of the granules. For native cornstarch, there is likely no mechanism for overcoming lubrication forces, as observed by an absence of shear thickening in cornstarch–ethanol suspensions.

**Discontinuous rate-stiffening in a granular composite**. An interparticle force scale is crucial in bringing about DST in various models[23,24], but it has been difficult to probe experimentally, as small-scale mechanical tests on particles in suspension are difficult to perform. To test the ubiquity of a force scale model, we fabricated a composite material composed of ~ 60–250 μm-diameter polydisperse fine sand coated with ~ 7.5% by volume cross-linked polydimethylsiloxane (PDMS) (see Fig. 4a for diagram and Methods for fabrication details). This PDMS-coated sand has a hard core provided by the sand particles, most of which is quartz, which has $E \sim 70$ GPa, and a soft ~ 2 μm-thick viscoelastic outer layer of PDMS, estimated from volume fraction. This composite sand has a force scale that originates from viscoelasticity rather than lubrication: at low deformation rates, loose sand grains can rearrange under small stresses, while at high deformation rates, the outer PDMS stiffens, bringing the sand grains into stable contact and temporarily forming a jammed network, with the hard particle cores providing a backbone for carrying the majority of the stress.

To show this, we performed impact experiments and flat-punch indentation tests. In impact experiments, bare sand is highly dissipative, absorbing the energy from the intruder upon impact and quickly stopping its progress. In contrast, the PDMS-coated sand responds elastically to the intruder, which rebounds several times after impact (Fig. 4b and see Supplementary Information for high-speed videos). This observation suggests that a viscoelastic coating on a stiff granular material imparts viscoelasticity to the overall composite through an IAS-like process where the grains jam under high deformation rates (~ 2.9 m/s). Mechanical flat-punch indentation experiments revealed

that the PDMS-coated sand has a nonlinear response to increasing displacement rates with a narrow range of displacement rates over which E increases by nearly an order of magnitude from 0.1 to 0.5 mm/s (Fig. 4c). This rate-dependence is absent in bare sand, where $E$ remains roughly unchanged for all deformation rates. The PDMS coating increases the $E$ of the coated sand at lower displacement rates, likely because of the interparticle cohesion introduced by the viscoelastic PDMS layer. The form of the rate-dependent response of the PDMS-coated sand (Fig. 4c) resembles that of the cornstarch and water mixture (Fig. 3), exhibiting a narrow range of displacement rates over which the $E$ increases rapidly. The response of PDMS-coated sand is also qualitatively different from that of native PDMS under indentation (Fig. 4d), which has a constant slope of ~ 0.8 when plotted in log($E$) vs. log(displacement rate).

We attribute the similarities between aqueous cornstarch suspensions and PDMS-coated sand to the presence of a force scale. In the case of cornstarch, the force scale is related to the swelling, which may serve to allow granules to overcome lubrication, and in the case of coated sand, the outer PDMS layer introduces a viscoelastic force scale, which modulates the quality of interparticle contacts, from soft and cohesive at low deformation rates to elastic and shear jammed at high deformation rates. We are unable to perform direct and accurate measurement of such a force scale at the particle level, but we can estimate the macroscopic stresses involved. For our cornstarch–water mixture, we estimate a macroscopic stress scale of ~ 20 Pa by measuring viscosity of the mixture under different shear rates (see Supplementary Figure 3 in Supplementary Information). This is in line with previous measurements for cornstarch and water[25,26]. For PDMS-coated sand, we estimate a critical force scale of ~ 4 μN in particle–particle contact. This value is estimated by calculating the contact force needed for bringing two approximately spherical sand cores (diameter ~ 155 μm) together (displacement of ~ 4 μm) with an effective modulus equal to the outer PDMS layer, which has an elastic modulus of ~ 100 kPa corresponding to a critical displacement rate of ~ 0.2 mm/s (Fig. 4c, d).

## Discussion

Interparticle contacts are essential for the emergence of DST in simulations[22], but smooth hard particles are repelled from contact by the lubrication singularity. In theory, particle roughness can help to overcome lubrication and has a dramatic effect on the rheology[23,27], and in colloidal experiments, smooth particles are largely contactless, without DST, while rough particles readily form load-bearing contacts under shear[28]. There are also studies that suggest dangling polymers appear on cornstarch surfaces[29] and interparticle hydrogen bonding occurs[30]. Clearly, the quality of interparticle interactions is important for suspension rheology. Our results show that, for cornstarch, solvent chemistry influences granule mechanical properties, altering interparticle interactions and suspension rheology. Specifically, the DST and IAS observed in cornstarch suspensions is strongly linked to cornstarch swelling in water. In cornstarch and ethanol mixtures, we find no swelling and no DST or IAS. A number of models and theories have suggested the existence of a force scale that originates from interparticle lubrication[14,23,24]. We postulate that cornstarch swelling is essential for the emergence of this force scale, which is responsible for the unique rheological properties of cornstarch in water. In other suspensions, a force scale may emerge from alternative sources. For example, stabilizing coatings such as steric layers in colloids impart a soft polymer outer layer, and such coatings are often essential for making stable, dispersed suspensions.

Expanding on the idea of an interparticle force scale, we artificially introduced a force scale in sand by coating it with a thin ~ 2 μm viscoelastic PDMS layer, which imparted a DST/IAS-like rate-dependence to the otherwise dissipative sand. Our composite granular material is qualitatively different from both native PDMS and native sand: (1) the stiffness of PDMS-coated sand is systematically higher than native PDMS and sand across all deformation rates, likely due to interparticle cohesion, and (2) the stiffness of PDMS-coated sand exhibits a sharply increasing region, i.e., from 0.1 to 0.5 mm/s in displacement rate, which is absent in both PDMS and bare sand. Future deformation rate-stiffening materials may utilize this insight for improving on flexible materials that respond dramatically to impacts. For example, the motif of a stiff granular material coated with a thin viscoelastic material can lead to flexible body armors that do not need density matching, steric stabilization, or suspending fluids, freeing up limitations on the properties of available solutes and solvents and significantly reducing the overall weight. One such example would be aluminum or diamond powder coated with PDMS for a lightweight, flexible, yet tough and highly rate-stiffening material.

## Methods

**Water absorption measurement**. For measuring the correct packing fraction of our cornstarch and water mixture that incorporates both effects of cornstarch swelling and water absorption by cornstarch, we first measure the mass of cornstarch and calculate its volume with the density of cornstarch taken to be $\rho_c = 1.59$ g cm$^{-3}$, as reported by Brown et al.[2]. We then prepare water of ~ 1.5 times the volume of dry cornstarch in a graduated cylinder to ensure a well-mixed state (see Supplementary Table 1 and Supplementary Note 2). The total volume change is calculated by comparing the measured volume, $V_m$, to the sum of the water volume and dry cornstarch volume, $V_w + V_c$. From this comparison, we found an ~ 11% increase in $V_c$, i.e., $V_m = V_w + 1.11 V_c$, as $V_w$ remains constant. This coupled with our direct measurements of cornstarch swelling suggests that cornstarch absorbs water during the swelling process, resulting in an overall ~ 11% increase in cornstarch volume.

For a thought experiment, suppose our null hypothesis is that no swelling occurs. The cornstarch density under that hypothesis would be $\rho_c/1.11 = 1.43$ g cm$^{-3}$. Since this density value is well outside of the range of measured values for cornstarch density (1.5–1.62 g cm$^{-3}$)[2,14,16,17], we reject the null hypothesis. This is another observation that supports the finding that cornstarch volume increases from adding water.

**Suspension impact experiments**. We dropped a metal disc from varying heights into a cornstarch suspension (water and ethanol) with packing fractions $\varphi_{nom}$ ~ 45% (the packing fraction is calculated with a cornstarch bulk density of 1.59 g cm$^{-3}$). We guided the disk with a chute located above the container, which has photoelastic gelatin boundaries. We used these boundaries to track the force propagation during impact (see Supplemental Materials). The disk had a diameter of 63.5 mm, width 11 mm, and mass 291 g. We recorded impacts with a Photron FAST-CAM SA5. We tracked the impactor using a circular Hough transform at each video frame, and numerically computed the velocity (refer to Supplementary Information for additional information and photoelastic data).

**Flat-punch indentation experiments**. Flat-punch indentation experiments were conducted with a TA instruments RSA III microstrain analyzer and indented with a 8-mm-cylindrical flat punch at two displacement rates, 0.05 and 4 mm/s. Packings of cornstarch in water and ethanol were made by adding solvent into granular cornstarch. In the case for water, we mixed the water and first allowed it to be absorbed. Experiments were conducted after the mixture settled into granular packing.

For flat-punch contact mechanics calculations, we used a standard rigid cylindrical indenter into flat plane scenario:

$$F = 2aE^* d, \qquad (1)$$

where $F$ is the normal force, $a$ is the cylinder radius (4 mm), $d$ is the indentation depth, and $E \sim E^*3/4$, assuming $v \sim 0.5$ and a very stiff cylinder (brass).

For sand particle–particle contact force scale estimate, we used a contact between two spheres:

$$F = \frac{4}{3} E^* R^{1/2} d^{3/2}, \qquad (2)$$

where $F$ is the normal force, $E \sim E^*3/2$, $R$ is the effective radius (1/2 the particle radius), and $d$ is the displacement.

**Coating PDMS on sand**. Polydimethylsiloxane-coated sand was prepared by first desiccating and heating fine sand at ~ 180 °C. Afterward, 7.5% by bulk volume hydroxyl-terminated PDMS was introduced and mixed with the sand until the sand was uniformly coated. We pre-dissolved boric acid in ethanol and introduced an ~ 0.5% by bulk volume (of boric acid) into the PDMS-coated sand, mixing the dissolved boric acid as the ethanol solvent evaporates. This concentration is not crucial—it is only important to include enough boric acid to crosslink all of the PDMS. After the PDMS cross-links and the mixture cools, we introduced ~ 2% by PDMS volume of oleic acid to act as a plasticizer for PDMS.

**Fabrication of native PDMS**. Native PDMS used in flat-punch test was prepared in a similar way. Approximately 6.7% of boric acid by bulk volume was introduced to hydroxyl-terminated PDMS, which was preheated to ~ 180 °C. After the PDMS cross-links and the mixture cools, we introduced ~ 2% by PDMS volume of oleic acid to act as a plasticizer.

## Code availability
The Code that support the findings of this study are available from the corresponding author upon reasonable request.

## Data availability
The data that support the findings of this study are available from the corresponding author upon reasonable request. The source data underlying Figs. 1e, 2a–b, 3 and 4b–d and Supplementary Figs. 2a–b and 3 are provided as a Source Data file.

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

## Acknowledgements
We thank Joshua A. Dijksman and Heinrich M. Jaeger for their insightful discussions. This work was supported by NSFC Grant No. 41672256 (H.Z.), NSF Grant No. DMR1206351 and DMR1809762, ARO Grant No. W911NF-18-1-0184, NASA Grant No. NNX15AD38G, the William M. Keck Foundation, a RT-MRSEC fellowship.

## Author contributions
H.Z. designed and performed the impact experiments with input from D.W. D.Z.C., H.Z. and D.W. designed and performed the rheology experiments. DZC fabricated the coated sand with help from D.W. D.Z.C., H.Z. and D.W. designed and performed the indentation experiments. D.Z.C., H.Z. and D.W. performed the microscopy experiments and analysis. R.P.B. supervised the experiments. D.Z.C. wrote the manuscript with input from R.P.B., H.Z. and D.W.

## Additional information

**Competing interests:** The authors declare no competing interests.

