## [Peer Review File · Nature Communications]

Reviewer #1 (Remarks to the Author):

In this article, the authors study shear thickening in cornstarch.

They postulate that shear thickening comes from the swelling of the grains and from the swelling of the grain. This swelling leads to the formation of an hard core and of a viscoelastic shell.

I have to major reservations dealing with the publication of this article.

First I am not convinced at all by the swelling of the grains.

Picture with higher magnification have to be taken and measurements on at least 100 particles have to be reported and averaged. The authors claim that the granule swelling increases the actual packing fraction of the cornstarch-water suspension by ~40%. This will correspond to a size variation of 12% and thus a variation of radius of 1 micron for particles of 10 microns. It is really difficult to see that on the pictures and on the movies.

Second the fact that the effective solid fraction is higher than estimated does not mean that the system is a shear thickening system.

There is for sure a link between high solid fraction and high viscosity. However this connection does not explain why the viscosity increases as a function of the shear rate.

In summary, this article deals with a study of cornstarch suspensions but does not bring any new convincing explanations to the shear thickening mechanisms.

The preparation of new shear thickening systems is interesting but their description is not suited for publication in Nature communication.

For these two reasons, I think that this article does not deserve publication in Nature Communication.

Reviewer #2 (Remarks to the Author):

This paper reports discontinuous shear thickening (DST) flow of dry fine rigid grains of sand that have been coated with a thin layer of soft elastomer. (Please state that the modified sand is studied dry, without a liquid medium.) This layer imparts sticky friction and soft viscoelastic cushion between the grains. The authors mention the viscoelasticity but they fail to point out the sticky friction. Both of these properties are likely very significant for the DST and impact activated solidification (IAS) behaviors of this particular granular material and should thus be discussed, tempering the conclusion that viscoelasticity is wholly responsible. The authors examine the swelling of cornstarch in water, which is found to be substantial. Little or no swelling is observed in ethanol. The results here are important at least in particular and it is unclear whether they are truly analogous to the DST IAS behaviors of cornstarch in water. To clearly establish the analogy, many more experiments would be necessary. Therefore the significance of this paper is to postulate and it would be improved if it also identified other work that would be suitable for testing the postulate. For example, the comparison with cornstarch and ethanol is incomplete because the two suspensions are not also compared at the same actual volume fraction. Such a comparison would help to identify the effects of particle surface properties.

Reviewer #3 (Remarks to the Author):

The authors presents the design, fabrication and characterisation of a novel shear thickening material with hard core and viscoelastic shell. The topic is interesting as it comes from normal two normal suspensions: cornstarch suspended in water and cornstarch suspended in ethanol. The dramatic difference in shear thickening effect for these two suspensions has inspired authors to investigate the viscoelastic effect, which is indeed reported very rarely. The original contribution is commended. Authors have conducted both experimental and theoretical works to verify the hypothesis, which sounds convincing. The work is recommended a publication provided the authors will address the following points:

(1) The viscoelastic effect consists of both elastic and viscous components, ranging from purely elastic to purely viscous. Are there recommendations of elastic component contributions for enhancing the shear thickening effect?

(2) The authors suggested certain new responsible materials with Aluminum powder coated with PDMS. Suppose similar experiments were conducted as reported in Fig. 4, what is the major difference between the new material results with those reported PDMS+Sand? Any criteria will be proposed to optimise the new materials?

Response to reviewers in blue
Reviewers' comments in black

List of major changes:

- 1) Provided supplemental data and statistics on measurements of swelling in over 100 cornstarch granules, as suggested by Reviewer #1. Supplemental images of sample measurements are provided in the supplemental materials.
- 2) Performed new experiments on packing fraction-matched suspensions of cornstarch+water and cornstarch+ethanol, accounting for cornstarch swelling in water, as suggested by Reviewer #2. These results are included in Figure 2 of the main manuscript, and additional description and discussion are also included in the main manuscript.
- 3) Performed new systematic experiments on the effect of indentation rate on the probed stiffness of the suspensions, showing dramatic differences between cornstarch+water and cornstarch+ethanol, as well as differences between low packing fraction (below $\phi \sim 0.54$) and high packing fraction (above $\phi \sim 0.54$) cornstarch+water suspensions.
- 4) Included additional discussion on the effect of granule-granule frictional interactions on the suspension rheology, as suggested by Reviewer #2.
- 5) Included additional discussion and conclusions on the effect of core material properties on the performance of a PDMS-coated granular composite, as suggested by Reviewer #3.
- 6) Added references and clarified the novelty of this manuscript in context with the current literature after discussion with Prof. Heinrich Jaeger (U Chicago).

Reviewer #1 (Remarks to the Author):

In this article, the authors study shear thickening in cornstarch. They postulate that shear thickening comes from the swelling of the grains and from the swelling of the grain. This swelling leads to the formation of an hard core and of a viscoelastic shell.

I have to major reservations dealing with the publication of this article.

We thank reviewer #1 for his/her time and effort in evaluating our manuscript. We are disappointed in the reviewer's conclusions of our work, and we have included a significant amount of additional data and analysis that we hope will convince the reviewer that 1) the cornstarch granules swell in water, and 2) there is *qualitatively* different mechanical behavior in cornstarch+water compared to cornstarch+ethanol, which we attribute to the swelling. We hope that, in light of this data and analysis, reviewer #1 will reconsider the significance and validity of our work.

First I am not convinced at all by the swelling of the grains.

Picture with higher magnification have to be taken and measurements on at least 100 particles have to be reported and averaged. The authors claim that the granule

swelling increases the actual packing fraction of the cornstarch-water suspension by ~40%. This will correspond to a size variation of 12% and thus a variation of radius of 1 micron for particles of 10 microns. It is really difficult to see that on the pictures and on the movies.

We apologize for the dearth of supplemental data to support our swelling findings. We have now included measurements of well over 100 particles as well as the corresponding images.

Second the fact that the effective solid fraction is higher than estimated does not mean that the system is a shear thickening system. There is for sure a link between high solid fraction and high viscosity. However this connection does not explain why the viscosity increases as a function of the shear rate.

We agree that the solid fraction does not dictate the shear thickening properties of the cornstarch system. Perhaps the reviewer was misled by our wording; to clarify, we postulate that the discontinuous shear thickening and impact-activated solidification is a result of swelling-induced changes in the cornstarch structure leading to qualitatively different mechanical properties in the cornstarch packings in water compared to in ethanol or dry. The increase in packing fraction owing to swelling serves to mechanically stabilize the packing (towards isostaticity), and is a prerequisite to the formation of force-bearing networks in response to shear and impact. To summarize, we postulate that viscosity increases dramatically with shear rate in cornstarch+water because cornstarch granule-granule interactions change, becoming 1) highly viscoelastic and 2) highly frictional after swelling, which does not appear in cornstarch+ethanol or in dry cornstarch. We attribute this change to the swelling that we observe in cornstarch+water, which also serves to increase packing fraction and mechanically stabilize the suspension, bringing the suspended granules closer to isostaticity.

In summary, this article deals with a study of cornstarch suspensions but does not bring any new convincing explanations to the shear thickening mechanisms. The preparation of new shear thickening systems is interesting but their description is not suited for publication in Nature communication.

For these two reasons, I think that this article does not deserve publication in Nature Communication.

Reviewer #2 (Remarks to the Author):

This paper reports discontinuous shear thickening (DST) flow of dry fine rigid grains of sand that have been coated with a thin layer of soft elastomer. (Please state that the modified sand is studied dry, without a liquid medium.) This layer imparts sticky friction and soft viscoelastic cushion between the grains. The authors mention the viscoelasticity but they fail to point out the sticky friction. Both of these properties are likely very significant for the DST and impact activated solidification (IAS) behaviors of this particular granular material and should thus be discussed, tempering the conclusion that viscoelasticity is wholly responsible. The authors examine the swelling of cornstarch in water, which is found to be substantial. Little or no swelling is observed in ethanol. The results here are important at least in particular and it is unclear whether they are truly analogous to the DST IAS behaviors of cornstarch in water. To clearly establish the analogy, many more experiments would be necessary. Therefore the significance of this paper is to postulate and it would be improved if it also identified other work that would be suitable for testing the postulate. For example, the comparison with cornstarch and ethanol is incomplete because the two suspensions are not also compared at the same actual volume fraction.

Such a comparison would help to identify the effects of particle surface properties.

We thank reviewer #2 for spending time and effort in reviewing our manuscript. We appreciate his/her constructive comments, and we have performed additional experiments comparing cornstarch+water and cornstarch+ethanol that we believe illustrate the dramatic and qualitative changes induced by the swelling of cornstarch in water on its rheological properties. We matched packing fractions ($\phi \sim 55\%$) in cornstarch+water and cornstarch+ethanol, taking into account the swelling-induced volume changes. In these ϕ -matched (after swelling) impact experiments, we observed that: during impact (fast time scales), 1) swelled cornstarch in water repels the intruder under impact, while unswelled cornstarch in ethanol merely stops the intruder, and after impact (slower time scales) 2) swelled cornstarch allows the intruder to sink to the bottom, while dense packings of cornstarch in ethanol arrest the intruder entirely. These observations suggest that impact into cornstarch+ethanol is qualitatively similar to impact into dense granular materials (the surrounding fluid does not contribute at low viscosities), whereas impact into cornstarch+water is qualitatively different: cornstarch+water repels the intruder, and the intruder sinks after impact, suggesting that the cornstarch resists impact (high speeds, short timescales) and deforms and flows around the intruder over low speeds and longer timescales. These dissipative and rate-insensitive results in ethanol and highly viscoelastic and rate-sensitive results in water strongly suggest that swelling is responsible for dramatic changes in the mechanical behavior of cornstarch granules. Additionally, our observation that the intruder sinks into the cornstarch+water suspension after impact may suggest that swelling leads to a softer, deformable outer layer on the cornstarch granules.

Reviewer #3 (Remarks to the Author):

The authors presents the design, fabrication and characterisation of a novel shear thickening material with hard core and viscoelastic shell.

The topic is interesting as it comes from normal two normal suspensions: cornstarch suspended in water and cornstarch suspended in ethanol. The dramatic difference in shear thickening effect for these two suspensions has inspired authors to investigate the viscoelastic effect, which is indeed reported very rarely. The original contribution is commended.

Authors have conducted both experimental and theoretical works to verify the hypothesis, which sounds convincing. The work is recommended a publication provided the authors will address the following points:

We thank this reviewer for his/her insightful and constructive comments.

(1) The viscoelastic effect consists of both elastic and viscous components, ranging from purely elastic to purely viscous. Are there recommendations of elastic component contributions for enhancing the shear thickening effect?

We believe the swelling-induced viscoelasticity qualitatively changes the cornstarch granule-granule interactions, enhancing the viscous and frictional contributions. On the other hand, the hard core dictates the elastic contributions, and we believe that its stiffness would contribute strongly to the shear thickening effect. So in short, we believe that the shear thickening effect would be amplified in engineered suspensions and coated granular materials with stiff cores.

(2) The authors suggested certain new responsible materials with Aluminum powder coated with PDMS. Suppose similar experiments were conducted as reported in Fig. 4, what is the major difference between the new material results with those reported PDMS+Sand? Any criteria will be proposed to optimise the new materials?

We believe that there are a number of important factors to consider for these rate-dependent coated granular materials: 1) the core material *stiffness* provides a backbone for the percolating force networks, which dictates the effective stiffness of the network upon impact. 2) The core material *yield strength* sets the upper limit on the impact velocity, as the material yields beyond a critical velocity, and 3) the coefficient of restitution, which depends on both modulus and yield strength, determines how much of the elastic energy from impact is stored.

Reviewer #2 (Remarks to the Author):

The additional data is certainly welcome (thank you) and the paper will eventually be suitable for publication. I request clarification of the following.

- The use of packing fraction ϕ is too confusing. There are two different uses in the paper, one is nominal, based on the separate densities of each component, and the other is actual, which is based on measurements of the swollen particles. Therefore, please use separate symbols. Perhaps use $\phi(\text{nom})$ whenever the actual and nominal packing fractions are different?
- The total volume change upon mixing cornstarch and water is said to be 11%. Is this expansion or contraction? How is it determined? The only description given is that a graduated cylinder was used. But how? Please report the density of the final solution (total mass/total final volume), and compare that to what would be expected if no volume change.
- I fear that the images in Figure 1 will be useless. It is already very difficult to see, at the current figure size. When this figure gets reduced in the journal, it will convey nothing.
- "In every case that we have accounted for" please specify here, for example a listing of the nominal packing fractions of your samples. This is on page 4, where "cornstarch in water reacts strongly to impact"
- Why is dry cornstarch data not shown?
- The description of the coated sand on page 8 is vague and unhelpful. Why is the particle core called a "backbone"? How does PDMS stiffen? This is very imprecise. What do you mean "PDMS layer carries the deformation"? Isn't it much more likely that the deformation at low rate is accomplished by granular flow? The ideas in this paragraph seem mixed up and connected improperly. The description "sticky" might also need elaboration.
- What is native pdms? This experiment needs to be described. Is native pdms in particle form? What are their size? How are they prepared? In Figure 3 d, how can they be so stiff? If the point of Figure 3 d is to show constant slope, why are there only four points?
- A force scale is usually quantified (in other papers). This paper however repeatedly uses the term force scale, without quantification.
- On page 10 the "coated sand exhibits a sharply increasing region"? What does region mean here? In what space are we talking?

The behavior of the dry but coated granular mixture does seem practical and thus important to publish.

Nature Communication MS# NCOMMS-18-00348A-Z / Chen *et al.*

Dear Dr. Elizaveta Dubrovina,

We thank the editors and reviewers for their time and effort in reviewing our manuscript. We find their comments very constructive and believe that we have made the necessary improvements that make this manuscript suitable for publication in *Nature Communications*. Here we include editor and reviewer comments (black) and our point-by-point responses (blue), and we have made all the changes in our manuscript (unmarked).

Sincerely,

David Z. Chen, Hu Zheng, Dong Wang, Robert P. Behringer

Comments from the editor

In an effort to ensure reproducibility of research data, we now also require that you provide a separate source data file. The source data file should, as a minimum, contain the raw data underlying all reported averages in graphs and charts, and uncropped versions of any gels or blots presented in the figures. To learn more about our motivation behind this policy, please see <https://www.nature.com/articles/s41467-018-06012-8>.

With in the source data file, each figure or table (in the main manuscript and in the Supplementary Information) containing relevant data should be represented by a single sheet in an Excel document, or a single .txt file or other file type in a zipped folder. Blot and gel images should be pasted in and labelled with the relevant panel and identifying information such as the antibody used. We also encourage you to include any other types of raw data that may be appropriate. An example source data file is available demonstrating the correct format: <https://www.nature.com/documents/ncomms-example-source-data.xlsx>

The file should be labelled 'Source Data', with the title and a brief description included in your cover letter, and should be mentioned in all relevant figure legends using the template text below:

"Source data are provided as a Source Data file."

We have attached our source data needed for reproducing every figure in our manuscript in the submission process.

Report of Review #2 – NCOMMS-18-00348A-Z

Reviewers' comments: Reviewer #2 (Remarks to the Author):

The additional data is certainly welcome (thank you) and the paper will eventually be suitable for publication. I request clarification of the following.

We thank the reviewer for the positive comments. Please see below our point-to-point responses.

- The use of packing fraction ϕ is too confusing. There are two different uses in the paper, one is nominal, based on the separate densities of each component, and the other is actual, which is based on measurements of the swollen particles. Therefore, please use separate symbols. Perhaps use $\phi(\text{nom})$ whenever the actual and nominal packing fractions are different?

We thank the reviewer for helping us identify this confusing aspect. We have specifically indicated which packing fraction we are referring to in our manuscript by using ϕ_{nom} for nominal packing fraction and ϕ_{act} for the actual packing fraction accounting for particle swelling.

- The total volume change upon mixing cornstarch and water is said to be 11%. Is this expansion or contraction? How is it determined? The only description given is that a graduated cylinder was used. But how? Please report the density of the final solution (total mass/total final volume), and compare that to what would be expected if no volume change.

We thank the reviewer for pointing this out and we should have been clearer. The total volume change upon mixing cornstarch and water is expansion. We found an ~11% increase in measured volume. For this volume change to be explained by an error in cornstarch density, the actual cornstarch density would need to be ~1.43 g/cm³. This is well outside of the range of typical values for cornstarch density (1.5 g/cm³~1.62 g/cm³). This is now described more in details in the main context and in the Methods session.

- I fear that the images in Figure 1 will be useless. It is already very difficult to see, at the current figure size. When this figure gets reduced in the journal, it will convey nothing.

We thank the reviewer for the comment, and we agree with the suggestion. We have zoomed in on a few particles to show the swelling in Figure 1. The full images are provided in the supplementary data source.

- “In every case that we have accounted for” please specify here, for example a listing of the nominal packing fractions of your samples. This is on page 4, where “cornstarch in water reacts strongly to impact”

We thank the reviewer for the suggestion. Specifically, “every case that we have accounted for” contains all the packing fractions for impact tests for water cornstarch mixture and ethanol cornstarch mixture. The range of packing fraction for these tests are $\phi(\text{nom})$ from 43% to 47% for water cornstarch mixture and $\phi(\text{nom})$ from 43% to 54% for ethanol cornstarch mixture. We have included these numbers in our manuscript.

- Why is dry cornstarch data not shown?

We didn’t include data for dry cornstarch for the flat-punch indentation test because for all the packing fractions tested for cornstarch+water, ethanol+water, sand, and PDMS+sand are either lower than or approximately random loose packing for dry cornstarch. Packing fractions lower than random loose packing cannot be realized in dry cornstarch, and dry cornstarch with above random loose packing does not produce forces that can be reliably measured by our instrument. This discussion now has also been added to our manuscript.

- The description of the coated sand on page 8 is vague and unhelpful. Why is the particle core called a “backbone”? How does PDMS stiffen? This is very imprecise. What do you mean “PDMS layer carries the deformation”? Isn’t it much more likely that the deformation at low rate is accomplished by granular flow? The ideas in this paragraph seem mixed up and connected improperly. The description “sticky” might also need elaboration.

PDMS coating is viscoelastic, hence stiffens as deformation rate increases. With this stiffening, a large amount of particles temporarily form a jammed network, hence with the hard particle cores providing a backbone for carrying the majority of the stress. It is more the collection of particle cores than a single particle core that forms the “backbone.” We have modified this part of discussion in our manuscript.

We agree with the reviewer that the deformation at low rate is accomplished by granular flow. By “PDMS layer carries the deformation,” we mean that the PDMS layer facilitates granular flow at low deformation rates while prohibiting it at higher rates by inducing granular jamming. We realize that this was unclear and have rephrased this discussion in our manuscript.

We agree that “sticky” is vague wording. The “stickiness” of PDMS comes from its viscoelasticity and self-cohesion, so we have removed that word in order to avoid confusion.

The discussion on page 8 now has also been rewritten and reorganized in our manuscript based on our response to this comment.

- What is native pdms? This experiment needs to be described. Is native pdms in particle form? What are their size? How are they prepared? In Figure 4 d, how can they be so stiff? If the point of Figure 4 d is to show constant slope, why are there only four points?

Native PDMS is prepared by the same method as that described for PDMS coated sand. Namely, hydroxyl-terminated PDMS is mixed with boric acid and then heated till PDMS cross links, after which ~2% by PDMS volume of Oleic acid is added as a plasticizer. The description for preparing native PDMS has been added in the Method session.

The obtained native PDMS is in a cross-linked solid-like form and is viscoelastic (solid on short timescales, flows on long timescales). The stiffness, i.e. elastic modulus, depends on the indentation displacement speed as shown in Figure 4d. One thing we would like to point out is that we made a mistake while generating Figure 4. The unit of y axis in Figure 4c and Figure 4d should be Pa, instead of kPa. This is a mistake when we formatted the figures. In addition, if the displacement speed is much lower than 0.01 mm/s, which is beyond the scope of our instrument, we expect the stiffness to be much lower than 10^4 Pa. The point of Figure 4d is more to show the rate dependence of the stiffness of native PDMS than to claim a constant slope. This rate dependence, happening in the same region of displacement rate tested in Figure 3c, supports our claim that the rate dependence observed in PDMS coated sand mainly comes from the property of PDMS coating.

- A force scale is usually quantified (in other papers). This paper however repeatedly uses the term force scale, without quantification.

We thank the referee for the comment. We have estimated the force scale to be ~20 Pa for the macroscopic critical shear stress for cornstarch and water mixture based on viscosity measurement. As for PDMS coated sand, we estimated the critical particle-particle force to be ~4 μ N from flat-punch indentation tests. These values have now been added in our manuscript, with details for obtaining/calculating the estimate.

- On page 10 the “coated sand exhibits a sharply increasing region”? What does region mean here? In what space are we talking?

We apologize for our ambiguity in this claim. By “a sharply increasing region,” we mean the displacement rate from 0.1 mm/s to 0.5 mm/s in the flat-punch indentation test. We have clarified this part in our manuscript.

The behavior of the dry but coated granular mixture does seem practical and thus important to publish.

We again thank the referee for the kind comments and constructive suggestions.

Reviewer #2 (Remarks to the Author):

Thank you for the edits.

- How do you conclude actual cornstarch density to be 1.43 g/cm³? Please spell out the density and volume calculations for me. In particular, please report the actual measurements, i.e. not approximations. Perhaps this could be included in supplemental material.

This request may seem to be associated with fine details. However, the swelling of cornstarch is a key issue here. And in the current state, the text is unclear and sloppy, and when I try to decipher it, I calculate an unphysical result, requiring more water than has been added. Please clarify this misunderstanding. It should be very easy to do so, and thus I think my request should be accommodated.

- The description of the pdms coating is inaccurate. Since that pdms is crosslinked, it does not behave fluid-like. A better description of its behavior is Figure 4d.
- Why is the force scale “artificial”? (caption of Fig 4.) Perhaps delete that word.

Reviewers' comments: Reviewer #2 (Remarks to the Author):

- How do you conclude actual cornstarch density to be 1.43 g/cm³? Please spell out the density and volume calculations for me. In particular, please report the actual measurements, i.e. not approximations. Perhaps this could be included in supplemental material.

This request may seem to be associated with fine details. However, the swelling of cornstarch is a key issue here. And in the current state, the text is unclear and sloppy, and when I try to decipher it, I calculate an unphysical result, requiring more water than has been added. Please clarify this misunderstanding. It should be very easy to do so, and thus I think my request should be accommodated.

The value 1.43 g/cm³ is not the actual cornstarch density that we wish to claim in our manuscript. This value is calculated as explained in the following logic. Upon well mixing cornstarch and water, we expect the total volume of the mixture to be $V_c + V_w$ if no swelling of cornstarch or absorbing of water in cornstarch occurs. Here V_c is calculated as the *pre-mixed* cornstarch mass (m_c) divided by the cornstarch density, $\rho_c \sim 1.59 \text{ g/cm}^3$, as reported by Brown, et al. [Phys Rev Lett 2009, 103, (8)]. After mixing cornstarch and water, the total volume of the *post-mixed* mixture was $1.11V_c + V_w$, assuming a constant V_w . If our null hypothesis is that no swelling occurs, the cornstarch density under the null hypothesis (no swelling) would be $m_c/(1.11V_c) =$

$\rho_c/1.11 = 1.43 \text{ g/cm}^3$. Because this is well beyond the range of cornstarch density reported in the literature, we reject the null hypothesis. In other words, the cornstarch swells (as we have measured from imaging) and also absorbs water. We conclude the water absorption occurs because this 11% increase in cornstarch volume is less than the total volume increase from swelling if no absorption occurs. We mentioned the value of $\rho_c/1.11 = 1.43 \text{ g/cm}^3$ merely to validate our observation of cornstarch swelling in water from another perspective. We have clarified this discussion in our revised manuscript.

As for the actual cornstarch density in the cornstarch+water mixture, there are two parts we need to take into account in our calculation: cornstarch swelling, which increases the cornstarch volume, and water absorption in cornstarch, which increases the total mass in cornstarch. For a given cornstarch volume V_c , the actual volume after swelling is $1.4V_c$ (12% increase in diameter). Assume the volume of water absorbed in cornstarch is $a*V_c$. Then from the volume measurement of cornstarch+water mixture mentioned above, we have the total volume V_t expressed in two ways: $V_t = 1.11V_c + V_w = 1.4V_c + (V_w - a*V_c)$. This gives $a = 0.29$. Hence, the actual cornstarch density in the cornstarch+water mixture is: $\rho = (\rho_c*V_c + \rho_w*a*V_c)/(1.4*V_c) = (1.59 + 0.29)/1.4 \text{ g/cm}^3 = 1.34 \text{ g/cm}^3$.

Three sets of experiments have been conducted and values in the volume measurement can be found below:

Cornstarch Mass (g)	Water Mass (g)	Cornstarch Volume Before Swelling (mL)	Water Volume (mL)	Measured Total Volume (mL)
200.22	301.91	125.92	301.91	440
188.76	200	118.72	200	335
242.04	204.64	152.23	204.64	371

We have included the calculation for the actual cornstarch density and data shown above in the supplemental information.

- The description of the pdms coating is inaccurate. Since that pdms is crosslinked, it does not behave fluid-like. A better description of its behavior is Figure 4d.

We thank the reviewer for pointing this out. The term fluid-like was intended to describe the PDMS-coated sand composite at low deformation rates. We incorrectly used it to describe

PDMS. Since this term is misleading, we have removed the “fluid-like” in describing the PDMS at slow deformation rates.

- Why is the force scale “artificial”? (caption of Fig 4.) Perhaps delete that word.

We thank the reviewer for his/her careful reading and pointing this out. The word “artificial” was originally used to suggest that the force scale is not natively present in either bare sand or PDMS. This description was misleading, and so we have deleted “artificial” in our manuscript.

Reviewer #2 (Remarks to the Author):

Excellent. The experimental information is very helpful, and it is important to include it. Thank you.
Please refer to the supplemental info in the methods section.